# Automatic Clustering of Indoor Area Features in Shopping Malls

**Ziren Gao** [1], **Yi Shen** [1], **Jingsong Ma** [1,2,*], **Jie Shen** [2,3,4] and **Jing Zheng** [1]

1    Department of Geographic Information Science, Nanjing University, Nanjing 210023, China
2    Jiangsu Center for Collaborative Innovation in Geographical Information Resource Development and Application, Nanjing 210023, China
3    School of Geography, Nanjing Normal University, Nanjing 210023, China
4    Key Laboratory of Virtual Geographic Environment (Nanjing Norma University), Ministry of Education, Nanjing 210023, China
*    Correspondence: majs@nju.edu.cn; Tel.: +86-1899-401-0172

**Abstract:** The comprehensive expression of indoor maps directly affects the visualization effect of the map and the user's map reading experience. Currently, only the points, lines, and polygons of outdoor maps are used as objects of cartographic generalization. Therefore, this study considers indoor map area features as generalization objects and deems the automatic clustering of the indoor area features of shopping malls as the research goal. The approach is used to construct an encoder-decoder clustering model, where the encoder consists of a graph convolutional network and its variant models. The results show that the proposed model framework effectively extracts the area features suitable for the indoor space clustering of shopping malls and improves clustering efficacy. Specifically, the model with the Relational Graph Convolutional Network as the encoder demonstrated the best performance, time complexity, and accuracy of clustering results, with accuracy up to 95%. This study extends the research object of cartographic generalization to indoor maps, enabling the automatic clustering of indoor area features, and proposes a clustering model for the important indoor scene of shopping malls. This is valuable for scholars interested in the cartographic generalization of indoor maps.

**Keywords:** indoor map; cartographic generalization; area features clustering; graph convolutional network

## 1. Introduction

Research on the cartographic generalization of indoor maps can facilitate multi-level expressions of indoor information and optimize the visualization efficacy of indoor maps and indoor navigation services. For shopping malls, the indoor map hierarchical expression and automatic comprehensive research may not only promote the informatization process of the malls and improve the level of refined management, but also provide customers with more humanized services. At present, the research objects of cartographic generalization are mainly outdoor maps, and multi-scale representations of indoor scenes and the cartographic generalization of indoor maps have received less attention. Compared to the outdoor space, the indoor space has a smaller scale, more compactly arranged internal elements, a more complex spatial structure, and a richer semantic information. As the expression of indoor maps has only one level, if the scale suitable for outdoor map expression is directly applied to the indoor map, the elements would be crowded, and the expression would not be clear. Wu et al. [1] noted that cartographic generalization could surpass its initial connotation and be gradually extended to indoor maps, virtual reality, volunteered geographic information crowd-source maps, robot maps, and other types. Gotlib and Marciniak [2] proposed that the problems to be considered in indoor map representation include the cartographic generalization of indoor maps.

Among the few studies on cartographic generalization of indoor maps, Jiang [3] explored multi-scale visualization strategies for point-of-interest point elements in indoor maps. Normann and Njaerheim [4] combined level-of-detail technology [5] and the cartographic generalization theory to explore the expression of indoor spatial information and improve users' image reading experience. However, that study did not consider the characteristics of indoor area features other than connectivity, nor did it explore the automatic generalization of indoor area features.

Area features occupy the largest area of an indoor map; therefore, the generalization of area features is especially critical in the cartographic generalization of an indoor map. In the existing generalization research on area features, the merging of area features is the core tenet of area features generalization, and the grouping of area features is the premise of merging [6]. Grouping determines the features that should be merged [7], which is called area feature clustering. The core tenet of clustering research focuses on measuring the similarity between features and the choice of clustering algorithm.

Research that measures the similarity of indoor area features is divided into two aspects: spatial relationships and attribute characteristics. Spatial relationships include spatial proximity and topological adjacency, in which the existence of edges indicates that nodes are adjacent to each other in space and the type of edges indicates the topological adjacency of area features. Attribute characteristics include geometric and semantic characteristics. Geometric characteristics are represented by the three directions of size, direction, and shape. Perimeter and area are common indicators for measuring the sizes of area features. Longest Edge Orientation and Smallest Bounding Rectangle Orientation are indicators commonly used to measure the direction of area features. Smallest Bounding Rectangle, Equal Area Circle, and compactness based on a square relationship are indicators commonly used to measure the shapes of area features. Semantic characteristics are major features of indoor area features that differ from their outdoor counterparts. As the ground features expressed by indoor maps usually have a strong functionality, semantic characteristics are highly important in clustering indoor area features. At present, there are several studies on the semantic characteristics classification of indoor features in academia. Deng et al. [8] selected and simplified features based on the characteristics of indoor space and the expression principles of simplicity and hierarchy of indoor maps and ultimately determined them to constitute frame background, key, and interest features. Ying et al. [9] used the shopping mall as an example and divided interior spaces into three categories: basic features represented by toilets and rest areas, shopping features represented by shops and cashiers, and indoor passage features represented by entrances, exits, and stairs. This study agrees and adopts this classification method.

Commonly used clustering algorithms include partition [10], hierarchical [11], and neural network [12–14] clustering algorithms. The partition clustering algorithm and the hierarchical clustering algorithm require considerable manual calculation and cannot provide automatic clustering of area features. Human participation can also add subjectivity to research results and increase the inefficiency of clustering work. Therefore, for indoor maps with many area features, these two methods are unlikely to achieve a good clustering effect. These limitations led to the emergence of the neural network clustering algorithm, which extracts high-level characteristics through multi-level characteristic learning and improves the effectiveness of characteristic extraction. Applying artificial intelligence to clustering tasks also provides automatic clustering to a certain extent [15]. Owing to the complexity of the indoor area feature and the clustering problem, existing neural network clustering methods still have considerable room for improvement. For example, as general neural network methods cannot process vector data, Graph Convolutional Networks (GCNs) [16–18], which process graph data, have attracted the attention of scholars.

In some studies related to cartographic generalization, Zhang et al. [19] used GCNs to automatically select road networks; Yu et al. [20] used GCNs to facilitate shape recognition and the classification of resident area features, and Ling et al. [21] used GCNs to enable

the recognition of building patterns. This study uses GCNs to extract characteristics of shopping mall indoor area features to achieve subsequent clustering.

The neural network clustering algorithm divides the entire clustering process of indoor area features into two steps [22]. First, the neural network model is used to represent the original indoor high-dimensional data as low-dimensional features that are more suitable for cluster analysis, after which the low-dimensional feature vectors are processed through the clustering operation. The most commonly used models are the autoencoder [23,24], generative adversarial network [25], variational autoencoder [26,27], and graph neural network models [28].

In the characteristic clustering process, the choice of clustering model often depends on which clustering concept is adopted. The current clustering concepts can be roughly divided into two types. These two clustering concepts correspond to two methods of deep learning: unsupervised and supervised learning. The supervised clustering algorithm combines the characteristic extraction process with the clustering task and adjusts the network parameters by minimizing the clustering loss, thereby assisting the neural network in extracting feature representations that are more suitable for the clustering task. To combine the supervised learning model and clustering problem of indoor area features, this study transformed the multi-classification problem of nodes into a binary classification problem of edges, according to Chang et al. [29], inspired by the Siamese network. The nodes with a high similarity have links and can be grouped into one category. Conversely, nodes with a low similarity have no links and do not belong to the same category. A Siamese network usually consists of two neural networks sharing weights; the network structure can be used to predict the similarity between pairs of samples, and then the difference between the calculated similarity and the label is defined as the loss function. The process of minimizing the value of loss function is the training process of the model. Chang et al. [29] used the Siamese network model to predict the similarity between input sample pairs and then labeled the sample pairs according to the similarity ranking. The value of the label is 0 or 1, where 0 denotes a low similarity and 1 represents a high similarity. The essence of this research is to continuously generate high-confidence positive and negative sample pairs based on the concept of self-learning and use them as supervised information to guide the training of the model [30].

The privacy of indoor space has led to restrictions on the development of indoor maps for many buildings, and shopping malls have become the most widely used scene due to their large area, complex structure, and openness. This study considers the automatic clustering of indoor area features of shopping malls as the research goal, constructs an encoder-decoder model based on GCNs, and compares and analyzes the applicability of different GCNs models to the research on automatic clustering of indoor area features to find the most suitable method for the automatic clustering of indoor area features.

A summary of the research goals and contributions is as follows:

(1) Different indoor maps have varying characteristics and uses, and the basis for clustering also varies. Based on a comparison of spatial characteristics of indoor and outdoor maps and the existing clustering methods of outdoor area features, we propose a clustering scheme suitable for indoor area features of shopping malls, and divide the indoor clustering basis into geometric, semantic and topological characteristics.

(2) To perform automatic clustering of indoor area features, the neural network method is applied to the problem of area features clustering, and thus an encoder-decoder clustering model is constructed. The model consists of an encoder composed of GCNs and a decoder for similarity measurements.

(3) According to graph characteristics and node aggregation methods, the GCN model derives various variant models, and different graph convolution models have varying applicability levels to cartographic generalization tasks. To explore which model is more suitable for the clustering task of indoor area features, the Relational Graph Convolutional Network (RGCN) and Relational Graph Attention Network (RGAT) are utilized to extract characteristics of indoor area features, and the applicability of the

different GCN models to indoor clustering tasks are analyzed through a comparison of experiments and results.

## 2. Materials and Methods

### *2.1. Encoder-Decoder Clustering Model*

#### 2.1.1. Encoder

The traditional clustering algorithm cannot easily measure the similarity of indoor area features; however, the strong learning ability of GCNs enables them to perform deep characteristic extraction of graph structure data. Therefore, the encoder-decoder model based on GCNs was selected for characteristic extraction and clustering of indoor area features in this study. Owing to the unique topological adjacency characteristics of indoor area features, the encoder uses RGCN [31] and RGAT [32] to extract the spatial structure information and attribute information of indoor area features, whose core function is to facilitate the dimensionality reduction expression of node characteristics. GCN and the Graph Attention Network (GAT) [33] are included in the clustering experiments as controls.

RGCN performs aggregation operations on nodes of various relationship types and divides the graph structure into different sub-graphs according to the various attributes on the edges. Neighborhood nodes are aggregated on the sub-graph obtained by division, and then the aggregated results of each sub-graph are added and input into the next layer of the network. A common representation is as follows:

$$h_i^{(l+1)} = \sigma\left(\sum_{r \in R} \sum_{j \in N_i^r} \frac{1}{c_{i,r}} W_r^{(l)} h_j^{(l)} + W_0^{(l)} h_i^{(l)}\right) \tag{1}$$

where $h_i^{(l+1)}$ is the characteristic representation of node $i$ at layer $l + 1$, $\sigma$ represents the activation function, $N_i^r$ represents the neighbor node index under relation $r \in R$, $C_{i,r}$ is the normalization constant, $h_i^l$ represents the state of neighbor node $j$ at layer $l$, $W_0^l h_i^l$ represents the learning of node $i$ itself, and $W_r^l$ is the weight matrix of relation $r$.

GAT can learn the weight coefficients between nodes by itself, which introduces the attention mechanism to GCN to prevent GCN defaulting to all neighboring nodes affecting the same weight. RGAT extends the attention mechanism to the RGCN model, and the calculation formula is as follows:

$$h_i^{(l+1)} = \sigma\left(\sum_{r \in R} \sum_{j \in N_i^r} \alpha_{i,j}^{(r)} g_j^{(r)}\right) \tag{2}$$

where $\alpha_{i,j}^r$ is the attention coefficient generated for node $i$ with relation $r$, $j$ is a neighbor node of $i$, and $g_j^r$ is the intermediate eigenvector representation of node $i$ under relation $r$.

#### 2.1.2. Decoder

The decoder combines the characteristics extracted by the encoder into sample pairs and performs an inner product operation on the characteristic vectors of the sample pairs to obtain the similarity between the sample pairs. The link probability between them is then obtained through the Softmax function. The higher the link probability, the greater the probability that the sample pairs will be clustered into one class. Subsequently, an appropriate threshold is set to classify the probability into two categories, and only the links that exceed the threshold are retained, thereby dividing and clustering the entire indoor map structure.

This clustering concept adopts the supervised learning method, which selects reliable label information and loss functions. The sample pairs are labeled according to the similarity between them, where 1 denotes a high similarity and 0 represents a low similarity. The loss function is defined by the difference between the actual similarity and the label value of the sample, and the model is trained by minimizing the difference. As the clustering problem of points is converted into the classification problem of edges, the cross-entropy

loss function is more suitable for the proposed clustering model of indoor area features, as its prediction result is the corresponding probability value of one sample belonging to $n$ categories. The loss value describes the distance between two probability distributions, and the smaller the distance, the closer the two probabilities are. Its standard form is as follows:

$$L(y, \hat{y}) = \frac{1}{N} \sum_i -[y_i \times \log(\hat{y}_i) + (1 - y_i) \times \log(1 - \hat{y}_i)] \tag{3}$$

where $y_i$ represents the label of the sample $i$, $\hat{y}_i$ $i$ is the predicted value of the model, and $N$ represents the number of samples. The link relationship between area features is used as the label value $y$ in the cross-entropy loss function, and the output of the decoder is used as the predicted value $\hat{y}$.

*2.2. Model Building*

2.2.1. Model input

The input to the model includes the adjacency matrix, the characteristic matrix of points, and the characteristics matrix of edges. The adjacency matrix describes the connection relationship between nodes and reflects the spatial structure information of the indoor map. The characteristic matrix of points represents geometric and semantic characteristics, and the characteristic matrix of edges represents topological adjacency. The geometric characteristics include eight attributes such as the centroid coordinates of the indoor area features, the direction of the minimum circumscribed rectangle, and the extension degree. Its formula and definition are shown in Table 1. Semantic characteristics depend on functional attributes of indoor area features. This study roughly divides the area features of shopping malls into three categories: user interest features (e.g., shopping, dining, restrooms), traffic features (e.g., entrances, escalators, elevators), and non-open areas (e.g., electrical facilities, office areas), as shown in Table 2.

**Table 1.** Common indicators for measuring geometric characteristics of area features.

| Geometric Characteristic | Index | Calculation Formula |
|---|---|---|
| Size | Area | - |
| | Perimeter | - |
| Direction | Longest edge orientation | - |
| | Smallest bounding rectangle orientation | - |
| Shape | Compactness | $\frac{4\pi A_b}{P_b^2}$ |
| | Extensibility | $\frac{L_{sbr}}{W_{sbr}}$ |
| | Concavity | $\frac{A_b}{A_{ch}}$ |
| | Fractal degree | $1 - \frac{\log(A_b)}{2\log(P_b)}$ |

**Table 2.** Semantic classification of indoor area features in shopping mall.

| Semantic Feature Classification | Element Content |
|---|---|
| User Interest Elements | Shopping, leisure, dining, and restrooms, among others. |
| Traffic Elements | Entrance and exit, stairs, escalators, and elevators, among others. |
| Closed Data | Electricity facilities, and office areas, among others. |

2.2.2. Model Training and Parameter Design

The training of the model is to learn and adjust the parameters by calculating the derivative of the loss function with respect to each network parameter and thereby determine the optimal parameters of the model. The backpropagation algorithm is often used to train neural network models, which calculates the partial derivative of the loss function for the weight and bias of each layer in the network through the chain rule of derivatives [34]

and ultimately uses the gradient descent algorithm to update the parameters to reduce the error of the output layer. The process is usually divided into three steps:

(1)  Inputting the processed data and the initialization parameters of the model and performing the feedforward calculation layer-by-layer to obtain the net input and activation value of each layer until the link probability value of the last layer is output.

(2)  Calculating the error term for each layer in reverse. For a neuron in layer $l$, its error term is equal to the weight sum of the error terms of all neurons in layer $l + 1$ connected to this neuron.

(3)  Updating the model parameters according to the gradient descent method to minimize the error. The specific process is to calculate the gradient of the loss function with respect to the weight and bias of each layer. In the gradient descent method, the weight parameters of the model are updated by the following formula:

$$W = W - \eta \frac{\partial L(y, \hat{y})}{\partial W} \tag{4}$$

where $\eta$ represents the step size of each update, which is called the learning rate.

Some parameters in the model cannot be learned automatically and must be designed manually, which are called hyperparameters. The settings of hyperparameters are usually based on the experience of researchers and cannot be automatically updated through model training. The first selection of hyperparameters and subsequent adjustment for optimization are the focus of machine learning research. To reduce the time spent on parameter adjustments, this study uses a random search to adjust the hyperparameters in the model, wherein a fixed number of parameters based on a random strategy is used to optimize the model rather than attempting all parameter values. For the clustering model proposed in this study, the hyperparameters involved include an activation function, dropout rate, learning rate, epochs, batch size, and threshold, among others. The activation function of the model uses the ReLU (Rectified Linear Unit) function, which remains unchanged when the input is positive and effectively addresses the problem of gradient disappearance [19].

### 2.2.3. Evaluation Index of Clustering Results

First, the Area Under Curve (AUC) is used to evaluate the clustering model performance. The output result of the clustering model is the link status between nodes. If there is a link, it belongs to a positive sample, and if there is no link, it belongs to a negative sample. For nth samples sorted from small to large, the AUC calculation formula is as follows:

$$AUC = \frac{\sum\limits_{i \in D^+} rank_i - \frac{m(m+1)}{2}}{mn} \tag{5}$$

where $rank_i$ represents the sequence number of the $i$-th sample after sorting, and the value range is [1,N]; $D^+$ is the set of positive examples; and $m$ and $n$ are the number of positive and negative samples, respectively. The value range of AUC is [0,1]. The larger the value, the better the performance of the model.

Second, the Adjusted Rand Index (ARI) [35,36] is used to quantitatively evaluate the clustering results. ARI is an evaluation index for supervised clustering that operates by comparing the differences between the pre-labeled and experimentally obtained clustering results. The formula is as follows:

$$ARI = \frac{\sum\limits_{i,j} \binom{N_{ij}}{2} - \left[\sum\limits_{i} \binom{a_i}{2} \sum\limits_{j} \binom{b_j}{2}\right] / \binom{t}{2}}{\frac{1}{2}\left[\sum\limits_{i} \binom{a_i}{2} + \sum\limits_{j} \binom{b_j}{2}\right] - \left[\sum\limits_{i} \binom{a_i}{2} \sum\limits_{j} \binom{b_j}{2}\right] / \binom{t}{2}} \tag{6}$$

where $a_i$ and $b_i$ represent the number of area features contained in each category in the labeling and experimental results, respectively. $N_{i,j}$ represents the overlapping number of area feature in the results of annotations and experiments under the same category. The value range of ARI is $[-1,1]$, and the larger the value is, the stronger the clustering effect is.

### 2.3. Experiment

This study utilizes an indoor map of large shopping malls in Nanjing as the clustering object and collects indoor map data of 40 shopping malls on an AutoNavi map, which can contain as many as 20,000 area features. See Figure 1.

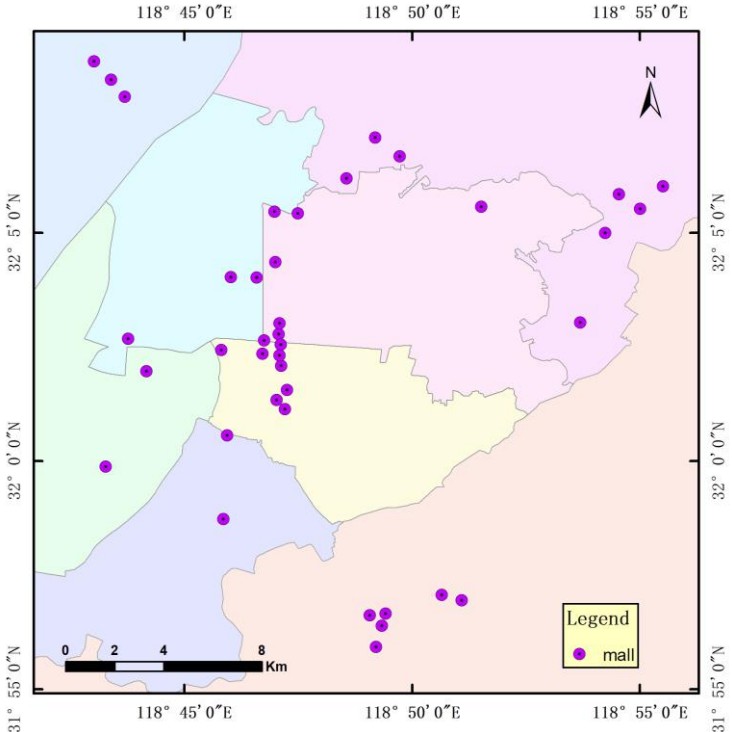

**Figure 1.** Data distribution map of shopping malls in Nanjing.

Before training the model, the sample data must be pre-processed, such as through vectorization, adding fields, and labeling. Subsequently, Python is used to convert the attribute field into the corresponding adjacency matrix and characteristics matrix as the input data of the model. The specific field processing process includes the following steps: (1) constructing the Delaunay triangulation according to the centroid coordinates of the area feature, and then converting it into the adjacency matrix of the graph, (2) numericizing and normalizing geometric characteristics and semantic characteristics to generate a characteristic matrix of nodes, and (3) generating a characteristic matrix of edges based on topological adjacency relationships between area features.

First, PyTorch Geometric (PyG) is applied, which is an extended library developed by PyTorch for deep learning on graphs. It was used to build the model, which supports building data as a graph and as a direct input into the model. Second, the samples are divided into training and test sets in a ratio of 8:2 using a random stratification method. Each sample includes 11-dimensional node characteristics representing 11 attribute fields of area features. Thus, the number of channels in the input layer to the model is 11. In the experiment, the number of channels in the hidden layer is 64, and that of the output layer is 2, indicating that the link either exists or does not exist, respectively. Third, the parameters and hyperparameters of the model are initialized and then fed into the samples for training. The key to model training is to use the loss function to calculate the difference between the actual predicted value and the expected output value and to minimize the loss value

by adjusting the hyperparameters to make the model reach a state of convergence. After repeated loss calculation and parameter updates, the dropout rate of the model is finally determined as 0.5, the batch size is 16, and the number of epochs is 500. Other specific parameters are shown in Table 3.

**Table 3.** Parameters of encoder model.

| Serial Number | Parameter | Numerical Value |
|:---:|:---:|:---:|
| 1 | Hidden size | 64 |
| 2 | Num layers | 2 |
| 3 | Num features | 11 |
| 4 | Batch size | 16 |
| 5 | Dropout | 0.5 |
| 6 | Learning rate | 0.001 |
| 7 | Epochs | 500 |

## 3. Discussion

### 3.1. Performance Evaluation

We recorded the changes in AUC and the loss value of the epochs during the training of four clustering models to evaluate the performance and training effect of each clustering model. As demonstrated in Figure 2, the overall change trends of the models are roughly the same: (1) loss shows a downward trend, and the value begins to stabilize after 100 iterations. RGAT and RGCN have similar losses, which are lower than that of GAT and GCN. (2) AUC values show an upward trend and then stabilize after 100 iterations. This trend indicates that the encoder-decoder clustering framework can effectively learn characteristics of area features suitable for clustering. In addition, the performance rates of RGAT and RGCN are higher than those of GAT and GCN, which indicates that RGCNs are more suitable for characteristic learning of indoor area features.

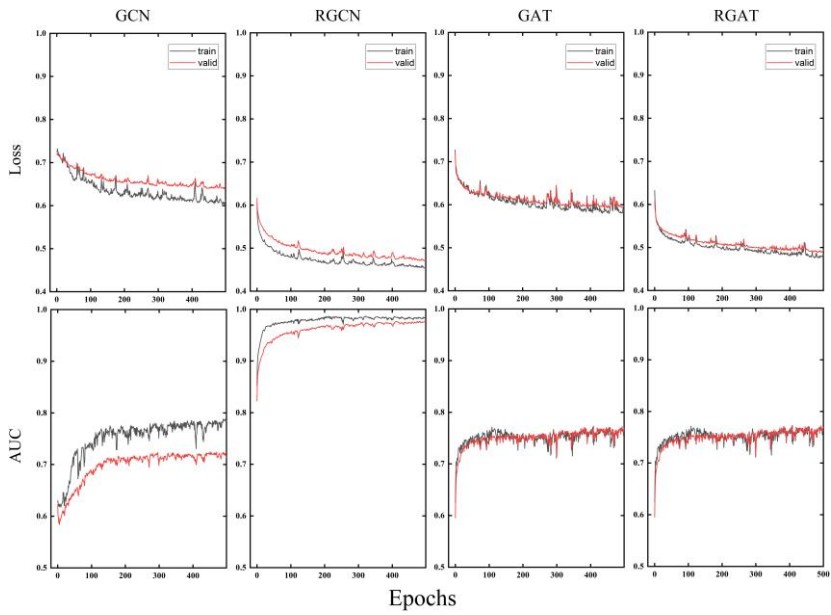

**Figure 2.** Loss and AUC of the four studied models varied by iteration.

The prediction time of the model reflects the time complexity of the model in predicting a sample, which is often used to measure the performance of the model. The prediction time of each model is shown in Figure 3. RGCN divides the entire graph structure into different subgraphs according to the relationship type and ultimately aggregates them separately; therefore, the time complexity is slightly higher than GCN. GAT adopts an attention mechanism and calculates the attention coefficient for each node separately before

aggregation; hence its model design is much more complicated than the GCN model. The reason why the time complexity of the GAT model is higher than RGAT is that the calculation of the attention coefficient of RGAT is simpler for each node after distinguishing the relationship types, but the attention coefficient value of GAT considers all values of neighboring nodes.

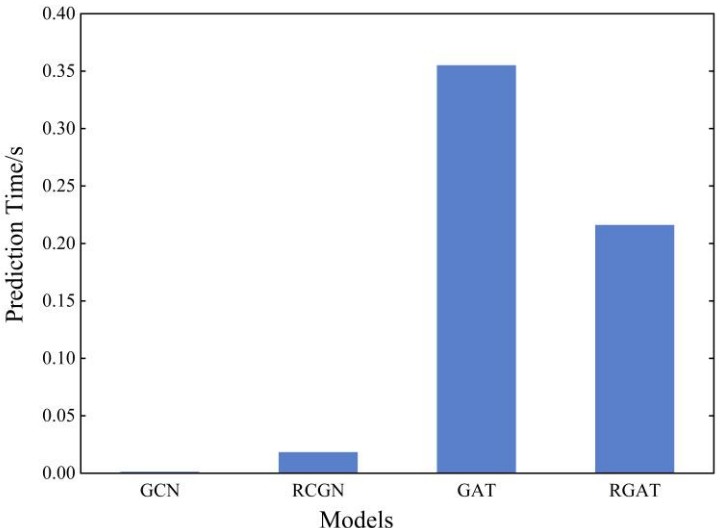

**Figure 3.** Prediction times for different models.

### 3.2. Analysis of Clustering Results

The indoor map samples were predicted by the trained clustering model, and the ARI values of the prediction results of the four models were obtained, as shown in Figure 4, which quantitatively describes the clustering efficacy of each model. The clustering accuracies of GCN and GAT in the control group are significantly lower than those of the experimental groups RGCN and RGAT, and the ARI is less than 0.4. The ARI of RGCN in the experimental group is the highest, close to 0.95, thus indicating that the expression of topological adjacency is highly important in evaluating the clustering of indoor area features. The unsatisfactory prediction results of the graph attention network likely occurred because the model was too complicated and not suitable for characteristic extraction of indoor area features.

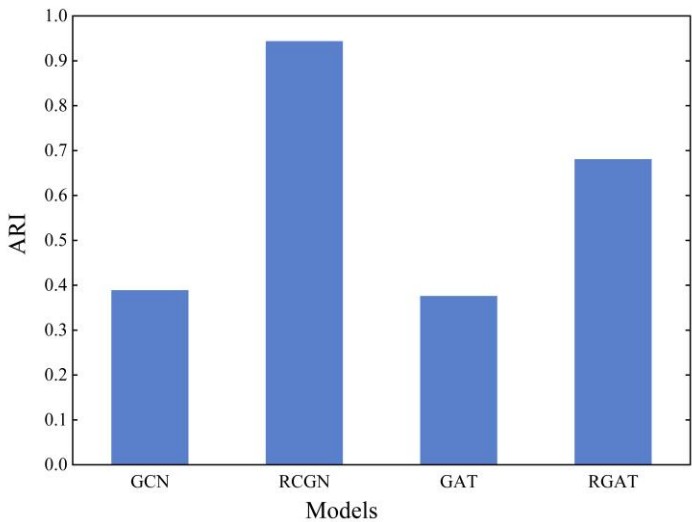

**Figure 4.** ARI values of the results predicted by the four models.

To further explore the clustering efficacy of the model and the influence of threshold parameters on the clustering results, RGCN was used as an example to record the changes in ARI with different threshold settings, as shown in Figure 5. The results show that the ideal clustering result in the partition clustering experiment was 0.62 as the partition value. If the link probability between two nodes exceeds 0.62, they are most likely to be clustered into one class.

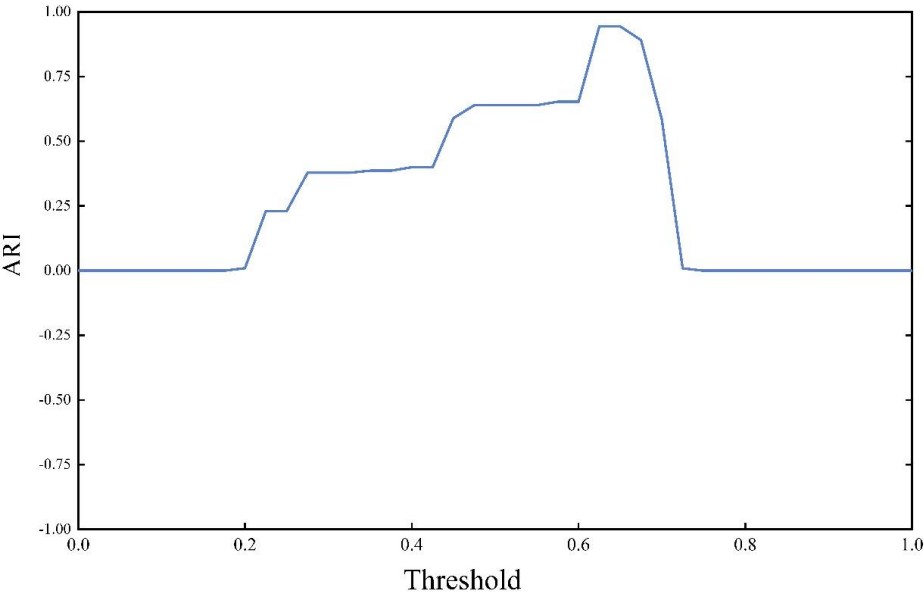

**Figure 5.** ARI values of the results predicted by the four models.

Combined with the above analysis and the data in Table 4, the RGCN model clearly demonstrated excellent results in terms of training loss, AUC, time complexity, and accuracy of clustering. To analyze the clustering efficacy of the model more intuitively, the trained RGCN clustering model was used to visualize different map samples. A map sample with a moderate number of area features was chosen. As shown in Figure 6, the left side represents the clustering result of the RGCN clustering model, and the right side shows the artificial clustering result. The area feature of the same color represents the same category. The same color in the two maps does not have any relationship, and color is just used to indicate which area features are merged. The overall effect of the model is clearly ideal. Although the spatial relationship and geometric shape of the samples are relatively complex, most of the area features that are connected and should be classified into one category demonstrate correct clustering, and only a small number of area features fail to cluster.

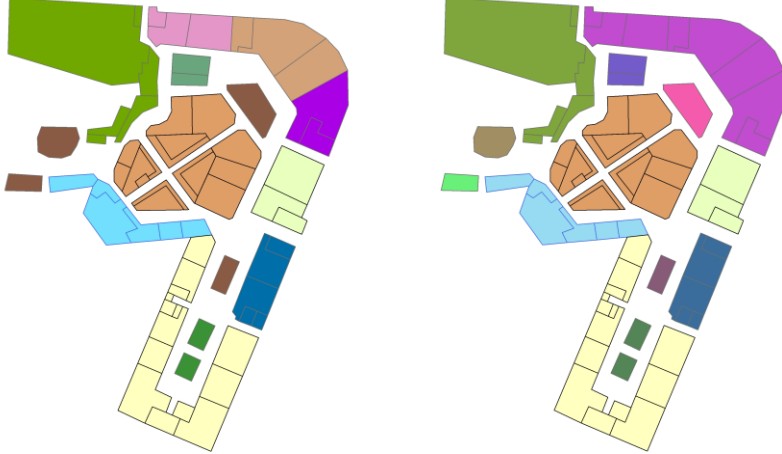

**Figure 6.** Clustering results of a sample (left: RGCN model, right: expected result).

**Table 4.** Time complexity and clustering accuracy of the four studied models.

| Models<br>Parameter | GCN | RGCN | GAT | RGAT |
|---|---|---|---|---|
| Prediction time/sec | 0.001 33 | 0.018 4 | 0.355 2 | 0.216 1 |
| ARI | 0.389 1 | 0.943 9 | 0.376 1 | 0.681 1 |

## 4. Conclusions

This study focuses on the cartographic generalization of indoor maps and uses deep learning to explore automatic clustering methods for indoor area features. We used an encoder-decoder model framework to complete deep clustering of indoor area features and applied GCNs to ensure automatic clustering of indoor area features. The encoders use RGCN and RGAT, and GCN and GAT are respectively used as control groups to compare and analyze the clustering effects of each model. First, the clustering scheme proposed in this study can achieve effective clustering of indoor area features either because of the accuracy of the clustering model or the visualization effect of the clustering results. Second, the proposed model has excellent clustering ability based on the ARI value and the clustering effect diagram. Therefore, the clustering model based on the encoder-decoder framework can effectively combine supervised learning and area feature clustering. Finally, through control experiments, RGCN has achieved the best results in model performance, time complexity, and clustering accuracy. Therefore, the RGCN model is highly suitable for indoor spatial feature extraction.

At present, the research on automatic clustering of indoor area features is still in the preliminary exploratory stage. Whether in terms of the clustering scheme, model construction, or network structure, automatic clustering should still be improved and optimized on a continual basis. Based on theoretical research and applied results, this study summarizes the following points for improvement.

*First: Optimizing the clustering scheme of indoor area features in combination with specific indoor scenes.* Compared to outdoor spaces, the indoor space has richer semantic characteristics and more personalized scenes; therefore, the principles of cartographic generalization are also different. This study only distinguishes semantic characteristics from the functional types of area features in the shopping mall and does not take into account the personal preferences of users, the purpose of using images and the specific characteristics of application scenarios. In future research, it is hoped that more scene characteristics and user needs will be considered.

*Second: Improving the availability of indoor maps and quality of clustering samples.* The data of the indoor map used in this study were collected, vectorized, and labeled manually. Hence, there are inevitable manual errors which will affect the experimental results to a certain extent. However, due to the current privacy of indoor data, it is difficult to directly obtain the vector data of indoor maps. Moreover, there are few cases for reference of the clustering of indoor area features. These factors have caused many limitations for the study of indoor spaces. It is hoped that there will be more ways to obtain the data of indoor maps in the future, and that there will also be more research on clustering principles of indoor maps and theoretical methods.

*Third: Using other deep clustering models.* This study attempts to describe the deep clustering method of indoor area features from the perspective of GCNs; however, in the field of deep learning, there are still many other available graph data clustering methods. In the future, various deep clustering models for graph data should be explored to improve the effectiveness and accuracy of clustering.

**Author Contributions:** Conceptualization, Ziren Gao; methodology, Ziren Gao, Yi Shen and Jingsong Ma; software, Jing Zheng; validation, Jing Zheng and Jingsong Ma; formal analysis, Ziren Gao; investigation, Jing Zheng; resources, Jie Shen; data curation, Yi Shen; writing—original draft preparation, Ziren Gao; writing—review and editing, Ziren Gao, Yi Shen and Jingsong Ma; visualization, Ziren Gao; supervision, Jingsong Ma; project administration, Jie Shen; funding acquisition, Jie Shen. All authors have read and agreed to the published version of the manuscript.

**Funding:** This research was funded by the National Natural Science Foundation of China, grant number 41871371.

**Institutional Review Board Statement:** Not applicable.

**Informed Consent Statement:** Not applicable.

**Data Availability Statement:** Not applicable.

**Acknowledgments:** Not applicable.

**Conflicts of Interest:** The authors declare no conflict of interest.

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
