# Peer review of "Automatic Clustering of Indoor Area Features in Shopping Malls"

_ijgi, doi:10.3390/ijgi12010019_

Round 1

Reviewer 1 Report

Overall, the structure and content of the paper are satisfactory.

Author Response

Thank you very much for your comments. There are specific changes to the manuscript in the attachment.

Reviewer 2 Report

Study on Indoor is very important. the spatial characteristic of outdoor has been studied by various scholars for a long time due to importance on human lives. the outdoor navigation is connecting human activity places to places. For this study performed, to generalize the indoor feature for enhancing indoor mapping and increase positive experience to users.

Authors many have difficulty to connecting their research method with the goal of the study. There are no clear conclusion and goal. What this study can make difference and impact on indoor experience. Authors also need to acknowledge that what is the main difficulty on understanding indoor space compare to outdoors based on both environmental and users perspectives. Authors need to very clearly address about why divide indoor spatial charactristic to three categories and importance on this study.

Author Response

Thank you very much for your comments, I have answered your questions below. There are specific changes to the manuscript in the attachment.

Quseton1Authors many have difficulty to connecting their research method with the goal of the study.

Response: When discussing research methods, we have added their relationship with study goal from paragraph five to paragraph eight of Section 1.

Quseton2There are no clear conclusion and goal. 

Response: We apologize that we did not describe the conclusions and goals clearly. Goals and contributions go hand in hand. Therefore, there are three goals. First, proposing a clustering scheme suitable for the indoor area features of shopping malls. Second, constructing an encoder-decoder model based on GCNs to accomplish automatic clustering of indoor area features of shopping malls. Third, exploring which model is more suitable for the clustering task of indoor area features. We have modified goals in the last paragraph of Section 1. The conclusion has been rewritten and discussed from three aspects, respectively, in the first paragraph of Section 4.

Quseton3 What this study can make difference and impact on indoor experience.

Response: Research on the cartographic generalization of indoor maps can facilitate multi-level expressions of indoor information and optimize the visualization efficacy of indoor maps and indoor navigation services. For shopping malls, the indoor map hierarchical expression and automatic comprehensive research may not only promote the informatization process of the malls and improve the level of refined management, but also provide customers with more humanized services. We have modified the first paragraph of Section 1.

Quseton4Authors also need to acknowledge that what is the main difficulty on understanding indoor space compare to outdoors based on both environmental and users perspectives. 

Response: Compared with the outdoor space, the indoor space has a smaller scale, more compactly arranged internal elements, more complex spatial structure, and a richer semantic information. We have modified the first paragraph of Section 1.

Quseton5Authors need to very clearly address about why divide indoor spatial charactristic to three categories and importance on this study.

Response: We agree with Deng et al. [8] that the three categories are sufficient to express the characteristics of indoor space and embody the principles of simplicity and hierarchical expression of indoor maps. Therefore, we choose to divide the interior space features into three categories. We have modified it in fourth paragraph of Section 1.

Reviewer 3 Report

The manuscript “ Automatic clustering of indoor area features in shopping malls” addresses a topic of interest to a broad audience and fits the journal's scope.

The comprehensive expression of the indoor map directly affects the visualization effect of the map and the user's map reading experience. This manuscript emphasises indoor map area features as generalization objects and deems the automatic clustering of the indoor area features of shopping malls. The result shows the model with the Relational Graph Convolutional Network as the encoder demonstrated the best performance, time complexity, and accuracy of clustering results, with accuracy of up to 95%.

The encoder-decoder model framework completes deep clustering of indoor area features and applies GCNs to ensure automatic clustering of indoor area features. This study collects indoor map data from 40 shopping malls to do the research and run proposed models. The clustering result shows a sample. Why did you choose this one? It would be best if you described the colour cluster representing what object.  (Page 12 Figure 6)

The conclusion makes me want to see your following study, not this one, so you should rewrite it.

Author Response

Thank you very much for your comments, I have answered your questions below. There are specific changes to the manuscript in the attachment.

Quseton1This study collects indoor map data from 40 shopping malls to do the research and run proposed models. The clustering result shows a sample. Why did you choose this one? 

Response: There are many map samples. We choose a map sample with a moderate number of area features to display the clustering efficacy of the model intuitively. We have explained the situation in the last paragraph of Section 3.

Quseton2 It would be best if you described the colour cluster representing what object.  (Page 12 Figure 6)

 Response: Color is used to indicate that the area features are merged. Different colors do not mean different objects. We have explained this in the last paragraph of the Section 3.

Quseton3The conclusion makes me want to see your following study, not this one, so you should rewrite it.

Response: The conclusion has been rewritten and discussed from three aspects in the first paragraph of Section 4.

Round 2

Reviewer 2 Report

Thank you for your effort to enhance you paper. Your update conclusion is much more supportive for readers. If you are including the next step of your study would be great for the identify your expected achievement in indoor mapping. I think two hierarchical analysis based on the size of individual cluster or vertical clustering based on the different criteria with 2D clustering would be providing significant impact on effective indoor mapping or visualization.